# Metabolic Modelling as a Framework for Metabolomics Data Integration and Analysis

**DOI:** 10.3390/metabo10080303

**Published:** 2020-07-24

**Authors:** Svetlana Volkova, Marta R. A. Matos, Matthias Mattanovich, Igor Marín de Mas

**Affiliations:** The Novo Nordisk Foundation Center for Biosustainability, Technical University of Denmark, DK-2800 Kgs. Lyngby, Denmark; svevol@biosustain.dtu.dk (S.V.); mrama@biosustain.dtu.dk (M.R.A.M.); matmat@biosustain.dtu.dk (M.M.)

**Keywords:** metabolic modelling, data integration, metabolomics

## Abstract

Metabolic networks are regulated to ensure the dynamic adaptation of biochemical reaction fluxes to maintain cell homeostasis and optimal metabolic fitness in response to endogenous and exogenous perturbations. To this end, metabolism is tightly controlled by dynamic and intricate regulatory mechanisms involving allostery, enzyme abundance and post-translational modifications. The study of the molecular entities involved in these complex mechanisms has been boosted by the advent of high-throughput technologies. The so-called omics enable the quantification of the different molecular entities at different system layers, connecting the genotype with the phenotype. Therefore, the study of the overall behavior of a metabolic network and the omics data integration and analysis must be approached from a holistic perspective. Due to the close relationship between metabolism and cellular phenotype, metabolic modelling has emerged as a valuable tool to decipher the underlying mechanisms governing cell phenotype. Constraint-based modelling and kinetic modelling are among the most widely used methods to study cell metabolism at different scales, ranging from cells to tissues and organisms. These approaches enable integrating metabolomic data, among others, to enhance model predictive capabilities. In this review, we describe the current state of the art in metabolic modelling and discuss future perspectives and current challenges in the field.

## 1. Introduction

Cell metabolism represents all the biochemical reactions occurring within the cell. Those reactions provide the supply of precursors for biosynthetic purposes, energy for anabolic reactions, signaling metabolites and general maintenance processes. In order to maintain the cell homeostasis and ensure the availability of all the necessary components, metabolism is highly regulated by different mechanisms involving allostery, enzyme abundance, and post-translational modifications. Metabolic network complexity provides plasticity and robustness to the cell in response to perturbations, leading to a challenge for researchers to understand and predict the molecular mechanisms governing cell behavior.

Development of high-throughput technologies has provided researchers with the capability to analyze and quantify an increasing amount and variety of biochemical constituents in the living cells, the so-called omics data. These omics technologies enable the identification and quantification of biological molecules (metabolites, protein, mRNA, etc.) connecting the genotype with the phenotype. Due to the close relationship between metabolism and cellular phenotype, the development of the high-throughput technologies for metabolomics has been particularly important in the study of a large number of biological processes.

Thus, the increasing amount of data provided by high-throughput platforms together with the complexity of metabolic networks makes it implausible to assess the overall behavior of metabolism from the individual properties of its components. Therefore, the study of cell metabolism behavior as a whole must be approached from a holistic perspective in order to extract relevant biological knowledge from the increasing amount of omics data by instead the so-called data-driven approaches. 

In this sense, metabolomics in combination with computational methods has the potential to unravel how cell metabolism is modulated in response to endogenous or exogenous perturbations such as genetic mutations or changes in the environmental conditions. 

A number of methods have been developed to analyze the increasing amount of omics data, the so-called data-driven approaches. These methods are extremely useful to define patterns or correlations between metabolic species and/or between groups/conditions. However, these correlations do not necessarily imply causality, which makes it difficult, if not impossible, to infer specific molecular mechanisms associated with a given phenotype. In this sense, incorporating prior information, such as network structure, component interactions, or physico-chemical properties enhances the predictive capabilities of the analysis and enables the identification of the mechanisms governing the system’s behavior. This information is typically incorporated in the form of a computational model. Models can describe cellular functions organized into hierarchical levels that are tightly controlled by dynamic and intricate gene regulatory mechanisms, signaling pathways and metabolic control (e.g., feedback, feed-forward inhibition, and covalent modification of enzymes). The results of these regulations and the network topology make metabolic networks highly non-linear and complex systems. Thus, model-driven methods are required to cope with this complexity in order to unveil the molecular mechanisms responsible for changes in metabolite concentrations as well as the emergent properties (properties that cannot be inferred from the individual components of the system) of metabolic systems arising from its intrinsic structure.

Based on how the metabolic network is formalized, two main modelling approaches can be considered: (i) constraint-based approaches that incorporate the reactions’ stoichiometry and thermodynamic information to infer metabolic flux distributions and predict complex metabolic phenotypes; and (ii) kinetic models that simulate changes in metabolite concentrations over time by incorporating biochemical network stoichiometry, mechanistic reaction rate laws, kinetic parameters and enzyme concentrations. These approaches are complementary and can be used jointly to explain emergent properties, show hidden patterns, and build new hypotheses. In this review, we will describe the current state of the art of model-driven methods to integrate omics data, with special focus on metabolic data. Furthermore, we will look into the examples of metabolomics data integration, strengths, and limitations of such approaches, as well as the successful case studies. 

## 2. Metabolic Modelling Approaches

In brief, constraint-based modelling incorporates information about reaction stoichiometry and thermodynamics [1], while kinetic modelling requires adding kinetic parameters and regulatory mechanisms in addition to what is required in constraint-based modelling [2,3,4]. As a consequence, kinetic modelling precisely describes the dynamic behavior of a system. However, it is limited to relatively small-size metabolic networks due to the lack of kinetic parameters and computational challenges of integration of big sets of ordinary differential equations. At the same time, constraint-based modelling enables the analysis of large-scale metabolic systems at steady state. However, it cannot model the dynamic response of metabolism to a perturbation. In the following, we will briefly describe how metabolic networks are formalized via either kinetic modelling or constraint-based modelling (Figure 1).

### 2.1. Constraint-Based Modelling

Constraint-based modelling is one of the most widely used family of methods to study metabolism [1]. It allows researchers to understand the possible states of the system (i.e., cell metabolism) based on prior knowledge such as the occurring reactions, their substrates and products (i.e., metabolites), their stoichiometry and reversibility. Based on this information, a metabolic network is built and formalized as a stoichiometric matrix (S). In this matrix, each row represents a metabolite and each column corresponds to a reaction, while the entries represent the stoichiometric coefficient of a metabolite in a specific reaction. Metabolites consumed in a reaction have a negative stoichiometric coefficient, while metabolites produced by the reaction have a positive stoichiometric coefficient for that specific reaction.

Given a vector describing the flux of each reaction (v) and the stoichiometric matrix, one can determine how the metabolite concentration varies along time ($\frac{dx}{dt}$) by applying the following equation:(1)$$Sv=\frac{dx}{dt},$$
where the changes in metabolite concentrations over time ($\frac{dx}{dt}$) are given by the reaction fluxes and the system’s stoichiometric matrix. Yet, if we assume that the system is at steady state, i.e., the metabolite concentrations do not change over time, then we can describe the system as follows:(2)Sv=0

This ensures that the sum of fluxes for the reactions that produce a given metabolite is equal to the sum of fluxes that consume it, such that the metabolite concentration remains constant. In other words, there is no accumulation or depletion of intracellular metabolites over time. By solving this system of equations, it is possible to determine the flux distribution. The overall nutrient uptake and release are represented by exchange reactions, which describe the in- and out-fluxes that cross the boundaries of the system. Here, the system is usually defined as the cell, but can also be the cell and the media, depending on the modelling goals. These exchange reactions can be, for instance, metabolite uptake or secretion rates. Transport reactions describe the transport of a metabolite from one cellular compartment to another or between cell and media when the media is the part of the modelled system [5].

In addition to equality constraints ($Sv=\frac{dx}{dt}$), other kinds of constraints can be also added in the form of inequalities [6]. These are typically used to define lower and upper boundaries for reaction fluxes that can describe an enzyme’s maximum capability [7] or reversibility (reactions with a non-negative lower boundary are considered irreversible) [8]. These boundaries can be inferred from exometabolomics measurements and constrain the space of feasible flux solutions of the system. Constraint-based modelling formulations allow us to model large metabolic network systems without the need to specify kinetic parameters or enzyme/metabolite concentrations. However, because there are more variables (reaction fluxes) than equations (Sv=0), the described system is underdetermined, which leads to many possible solutions (flux distributions). A variety of methods has been proposed to find a solution space, i.e., a reduced set of feasible solutions, depending on the problem, some of which are described below.

Constraint-based modelling can be applied to problems of different scales. The applicability of this method to the genome scale is particularly important, since it provides an overview of the whole metabolism. With the development and wide usage of genome sequences, methods to automatically annotate the genome and to derive genome-scale models were created [9,10,11,12]. The process of annotation is based on so-called gene–protein–reaction rules (GPRs), using information that links enzymes and protein identifiers, reaction names and reacting metabolites and more recently stoichiometric GPRs (S-GPRs) that define the number of transcripts required to generate a catalytically active unit (enzyme) [13]. After automatic annotation and manual curation, a draft of the plausible metabolic capabilities of the cell is assembled. This significantly reduces the amount of time needed to create an organism-specific model.

Flux balance analysis (FBA) [6,14,15] is one of the most widely used constraint-based methods developed for simulating metabolism at the genome scale. FBA is a computational method that is usually used to predict flux distributions while optimizing a given cellular function, or a combination of them (i.e., maximize growth). FBA yields a specific solution, i.e., a flux distribution, where the objective function is guaranteed to be optimal. The idea that cells follow a specific functional objective, and if so which, is being debated, but thought to have an evolutionary reasoning [16,17,18,19]. One of the most widely used objective functions for actively dividing cells is the growth rate. In rich media, this approach is in good agreement with experimental data [20]. Other common optimization functions are the maximization of ATP produced, minimization of consumed glucose, or a complex combination of such functions. However, choosing an objective function describing a given cell phenotype is not always obvious, since it depends on many factors, for instance, on the type of cells modelled, the growth conditions or the availability of nutrients [18]. 

To explore alternative and suboptimal solutions as well as study the network flexibility, a variant of FBA, flux variability analysis (FVA) [16], can be used. FVA can determine a range of fluxes for each reaction such that the minimum growth rate should be at least 90% (or other specified percentage) of the maximum growth rate [21,22]. FVA is also applied to the metabolism of byproducts and secondary metabolites, where FBA solutions are unreliable [23].

While FBA with the objective of maximizing growth results in reasonable solutions for wild-type cells, it does not so for (unevolved) gene knockout mutants. While in principle wild-type cells optimize their growth, unevolved cells with knockouts do not. Since homeostasis governs metabolic reprogramming [24,25], we cannot assume that the cell will follow a common objective, such as maximizing its growth. To acknowledge the requirement for metabolic homeostasis, the minimization of metabolic adjustment (MOMA) approach was proposed [26]. The main idea behind this method is that, to maintain homeostasis, the difference in fluxes before and after the perturbation should be minimal. MOMA predicts the fluxes of a knockout strain by assuming that the cell will have a minimal redistribution of fluxes compared to its ancestor [26]. Another similar approach is the regulatory on/off minimization (ROOM), which minimizes the number of fluxes that are significantly different from the wild type [27]. Wild-type fluxes, determined by FBA or other methods, need to be known in order to use these approaches. 

For problems which cannot be addressed by traditional stoichiometric methods, such as —changing concentrations over time and/or changing steady states, —dynamic methods were proposed. The idea behind them is to solve multiple constraint-based problems, each at a different time point, assuming steady state at each of them, which allows the modelling of metabolic changes over time. This approach, called dynamic FBA (dFBA), was originally proposed by Varma and Palsson to predict the metabolism of *Escherichia. coli* throughout batch cultivations in different conditions [28]. 

In this review, we described constrained-based methods that are necessary to understand state-of-the-art approaches of metabolomics data integration. To obtain more information about the variability of approaches of constraint-based methods, we refer the readers to the review by Lewis et al. [29]. We also refer to other articles covering other powerful and widely used techniques such as the GECKO method [7], a method that enhances constraint-based methods to account for enzyme maximum capacity by incorporating kinetic constants, metabolism and expression models (ME models) [30] and protein allocation models [31,32,33,34].

A different way of approaching the analysis of metabolic networks is the elementary flux modes (EFM) analysis [35,36]. EFMs are minimal sets of reactions that lead to a steady state flux solution. Minimal refers here to the fact that they do not contain any loops in the metabolic network and that no reaction can be removed while still allowing for a steady state flux [37]. All the EFMs found in a EFM analysis (EFMA) for a given network describe the possible paths that can be taken through it, e.g., all the ways a substrate could take to be turned into a product under steady state conditions.

Among the major limitations of constraint-based models in their classical implementation is that they rely only on fluxes and stoichiometry and not metabolite concentration. Thus, under the steady state assumption, fluxes occur irrespective of the metabolite pools—so it is neither possible to incorporate nor predict concentrations in the system. The choice of objective function and the applicability of the steady state assumption have an important impact on the final solution [18]. Another limitation is that it does not have the ability to account for allosteric regulation, which depends on metabolite concentrations [38]. These limitations are addressed by imposing additional constraints or using different modelling approaches such as kinetic modelling [39].

### 2.2. Metabolic Flux Analysis

Metabolic flux analysis (MFA) [40] is a widely used tool for metabolic engineering, which has also become more common in other fields, such as biomedical applications [41] and basic science [42,43]. MFA follows the same principles as FBA but does not assume any functional objective, e.g., the growth rate or ATP production. Instead, MFA is an optimization problem that minimizes the difference between simulated and experimental flux data for constraint-based models (Figure 2). In other words, unlike FBA that uses metabolomics to define the boundaries of the system and constrain the space of feasible flux solution, MFA incorporates the metabolomic data in the objective function in order to set a metabolic flux profile able to minimize the difference between the predicted and calculated secretion and uptake rates. There are two main types of MFA, differing on the use of isotopic labelling.

Stoichiometric MFA (stMFA) [44,45] does not require labelling experiments. It is used to balance the fluxes of extracellular and intracellular metabolites according to the biochemical network when given experimental measurements such as glucose uptake rate, CO_2_ secretion rate, or the growth rate. stMFA is based on minimizing the difference between the experimental and simulated secretion and uptake rates. Given stoichiometric constraints and measured fluxes, least square regression is performed. However, secretion and uptake rates are usually insufficient to resolve all the reaction fluxes in the model. Further, small-scale models are typically used, and therefore the reduction in genome-scale models is needed. Similar to dFBA [46], the dynamic extension of FBA, dynamic MFA, is also used to address changing concentrations in the environment.

Another technique is ^13^C-MFA [47,48], where isotope labelling patterns are used as additional constraints that help resolve the intracellular fluxes. ^13^C-MFA is based on minimizing the difference between the experimental and simulated secretion, uptake rates and isotopic-labelling pattern. This more detailed metabolic information allows ^13^C-MFA to infer metabolic reaction fluxes, which otherwise could not be resolved, such as cycles, bifurcations or the source of NADPH production. Thus, when such a part of the metabolism is of interest, researchers should consider adding isotope labelling to their experiments. However, this implies more complexity in terms of experimental design and data analysis [40,49].

There are different ways to approach labeling experiments, depending on the molecules/reactions of interest and the required resolution. Methods range from measuring the labeling patterns of distinct sets of metabolites, e.g., central carbon metabolism intermediates [50], proteinogenic amino acids [51] or further bound entities such as glycogen-bound glucose or RNA-bound ribose [52], to increasing the analysis to the genome scale [53]. For more specific parts of metabolism that are of interest, subsets can be investigated, as described by Liu et al. for NAD metabolism [54].

It might not be possible to resolve the more complex metabolic architectures by a single labeling experiment and may require parallel setups with differentially labeled tracers, for example, a mixture of different tracers of a single substrate, such as (1-^13^C)glucose and (^13^C_6_)glucose, or different labelled substrates, such as (^13^C_6_)glucose and (^13^C_6_)glutamine [55,56].

While measuring the labeling at metabolic and isotopic steady state is the commonly used approach, isotopically non-stationary metabolic flux analysis (INST-MFA) harbors distinct advantages in certain aspects. For instance, by taking time-dependent labeling of metabolites into account, metabolite pool sizes and reversible exchange fluxes can be estimated. This allows for comparison with (or inclusion of) metabolomics data [8,57].

The above-described methods use the same formalism as constraint-based models: linear combinations of reaction fluxes and the steady state assumption. Notably, this approach (minimization of the difference between observed and modelled fluxes and isotopic pattern) can be applied in the kinetic modelling framework and such an application was demonstrated [58,59,60,61,62]. In this approach, kinetic parameters are refined and the flux distribution is determined. This approach was not widely explored, despite clear benefits: acquiring not only flux distribution that describes the metabolic phenotype, but also a kinetic model that has predictive and explanatory capacities.

For detailed information about MFA and a family of different MFA methods, we refer to this review [40] and protocol [52].

### 2.3. Kinetic Modelling

Kinetic modelling describes the changes in metabolite concentrations given an initial condition by representing the metabolic network as a system of differential equations, most commonly ordinary differential equations. The fate of each metabolite is described by an ordinary differential equation that consists of a rate equation for each reaction that either consumes or produces the metabolite. Solutions for such systems yield the concentration of metabolites at each time point, and make it possible to evaluate the evolution of fluxes and metabolites over time. The equations for the reaction rates are functions of the enzyme concentrations, kinetic parameters, and metabolite concentrations. Kinetic modelling describes not only the stoichiometric relation between substrate and products but also the enzymatic mechanisms and different levels of regulation (i.e., allostery or post-transcriptional regulation), which makes these methods suitable for the integration of multiple omics data types.

While constraint-based models are suitable to predict the state of the system, they do not answer how it is achieved. Further, it is not always possible to assume that the system is in a steady state. Since kinetic models describe the enzymatic mechanisms underlying a given metabolic reaction, this approach enables the incorporation of allosteric regulations, where the enzyme activity is affected by the binding of a small molecule to a site other than the enzyme’s active site [63]. Allosteric regulation [64] is an important component of cellular metabolism control and regulation [65,66,67]. Although possible [39], it is not straightforward to account for regulation in constraint-based models because this interaction typically depends on metabolite concentrations, which are not explicitly included in constraint-based models.

Kinetic modelling approaches have been widely applied in a variety of cases, such as inferring metabolite kinetics from experimental data [68,69,70,71], or developing the mechanistically accurate expressions to describe the involved reactions [72]. Due to insufficient availability and inaccuracy of kinetic parameters in literature, kinetic models are usually limited to studying smaller metabolic network systems [73]. Nonetheless, the feasibility of genome-scale kinetic models has been demonstrated [74,75]. 

Researchers are working on solutions to address current limitations, such as kinetic parameter paucity and small-scale preference. One possible approach to address the lack of kinetic parameters and/or uncertainty in the given kinetic parameters, and as a consequence in predictions, is to use ensemble modelling [76]. Among currently available ensemble modelling packages are approaches such as ORACLE [77], GRASP [78]. An update to the previous work is K-FIT [79], which significantly improves computational efficiency and makes it more accessible, enabling extending the use of kinetic modeling to larger network systems.

In ensemble modelling, the predictions of multiple models are combined in order to create a feasible solution space that encompasses all of them. The topology of a network and its steady state flux are used to infer the metabolic flux distributions. As much information as is available about the mechanism of enzymatic reactions in the model is included to keep the ensemble biologically feasible. The steady state flux can be determined experimentally, taken from literature, if available, or estimated based on the exchange fluxes of the system. An ensemble of models is created, spanning the range of thermodynamically allowed kinetic parameters, each reaching the same steady state flux and metabolite concentrations. Flux changes due to enzyme perturbation can be used to refine the analysis and reduce the number of models in the ensemble [76].

## 3. Metabolomics Integration

Numerous approaches have been developed to integrate metabolomics data into mechanistic models, such as kinetic or constraint-based models. These computational tools allow researchers to obtain new insights into metabolism regulation. From a mathematical point of view, we can describe two main types of data integration: variable constraints (i.e., fixing the upper and lower boundaries of a metabolic reaction flux or the relationship between two metabolic fluxes) and parameter fitting (i.e., kinetic parameters estimation by minimizing the difference between model predictions and experimental data). From a problem statement point of view, researchers can infer intracellular fluxes from the experimental data, estimate flux changes that lead to a specific phenotype, or find the parameters of the system (e.g., kinetic parameters such as the turnover rate or the Michaelis–Menten constant) to predict or describe another metabolic state (Figure 3). In this section, we will discuss different data integration examples (summarized in Table 1)—in particular, how problems are defined, types of data used and achievements.

### 3.1. ^13^C Metabolic Flux Analysis as an Isotope Labelling Data Integration

A major goal of metabolic research is to explain how metabolic networks are rewired and regulated in response to a given perturbation. Unlike proteins, transcripts, and metabolites, fluxes cannot be measured directly—they can only be estimated. Labelling experiments have been extensively used to determine flux values (relative or absolute) and pathway usage [80]. To acquire flux values, cells are cultivated using labeled substrates, e.g., glucose labeled with one or more stable ^13^C isotopes as a carbon source. The substrate is processed by the cell and converted into labeled products and intermediates. By measuring these products and intermediates, distinct labeling patterns are revealed, from which the fluxes through the reactions can be inferred [81]. The dynamics of the downstream metabolite labeling can also be measured directly and taken as the reaction fluxes [80]. ^13^C-MFA extends this idea to the systems biology approach. Together with previous knowledge about the cell metabolism, the data is integrated via ^13^C-MFA methods to infer a metabolic flux profile that is consistent with the experimental measurements. ^13^C-MFA integrates the prior information in the form of a network topology and carbon rearrangements to find the steady state that best satisfies the stoichiometric and mass-balance constraints, as well as the experimentally measured isotopic-labelling pattern. This is usually referred to as “fluxomics” in literature. ^13^C-MFA is now a gold standard in resolving flux distributions. ^13^C-MFA has been extensively used in different biological fields, such as metabolic engineering, cancer research, physiology, biochemistry, and plant sciences, both in prokaryotic and eukaryotic systems. 

^13^C-MFA was used in cancer cell culture research to show drug vulnerabilities and lead to possible treatment targets and combinations [82]. ^13^C-MFA was also applied to study cell metabolism in biotechnological processes. As an example, Pereira et al. used ^13^C-MFA to study the metabolic response of industrially relevant Chinese Hamster Ovary (CHO) cells to change the media composition in order to decrease the production of toxic ammonia as a byproduct of the fermentation [83]. Templeton et al. used ^13^C-MFA to identify metabolic reprogramming associated with high antibody productivity in CHO cells [84]. Aguilar and colleagues studied metabolic reprogramming associated with cancer stem cells in a dual cell model consisting of one highly metastatic subpopulation displaying cancer stem cell properties and a second non-metastatic subpopulation displaying a stable epithelial–mesenchymal transition [58]. Experiments with the incorporation of ^13^C-glucose were used to estimate the flux distribution and showed differences in cellular respiration between those two cell types. In addition, model-driven strategies were used to support other findings in a study that suggest differences between metabolic features of cancer stem cells and epithelial–mesenchymal transition phenotypes. Each of the phenotypes can be linked to optimal cell proliferation and sustaining specific cell phenotypes. This study provides an example of how isotope-labelling data can be used with kinetic models to determine the intracellular fluxes. Labeling metabolomics in combination with kinetic modelling can also be used to infer the intracellular metabolic network topology in order to study physico-chemical phenomena such as channeling [85] or crowding [86]. However, due to the paucity of atom transition information and kinetic parameters, this approach is limited to very well-known metabolic systems, which limits the scope of this approach and the number of available software solutions.

While in general the models used in ^13^C-MFA are small scale, genome-scale models have been introduced recently. One of such approaches was introduced by Martin and coworkers [87] who proposed a method to constrain a genome-scale model with ^13^C labeling data. They assumed that the flux flows from the central carbon metabolism to the peripheral parts of metabolism and does not flow back. ^13^C-MFA is then used to infer the flux distribution in central carbon metabolism, which is used as a constraint to predict the flux distribution in peripheral metabolism. One advantage of this approach is that it does not require any strong evolutionary or physiological assumption, such as the maximization of the cell growth rate or ATP production. Another example is the work by Gopalakrishnan and Maranas [88], where they demonstrated the possibility of extending ^13^C-MFA to a genome-scale model. This study highlighted the variation in flux values that is introduced by selecting models of different sizes. The authors point out that even though using genome-scale models should be preferable, it is impractical due to the increase in computational time associated with a higher number of variables. McCloskey et al. [53] used ^13^C-labelled metabolomics to constrain a model that describes 537 reactions and 429 metabolites, and showed a significant increase in the number of calculated fluxes. This included measurements for metabolic intermediates in central carbon metabolism and cofactors of peripheral metabolism. The authors also used this approach to study the evolution of gene knockout strains of *E. coli* and find how the cells adapt to gene loss [89]. 

^13^C-MFA is proven to be a valuable approach to determine flux distributions. However, it still remains a challenge for many systems, especially those with slow labelling dynamics, multiple compartments, and different carbon sources. An assessment of the measurement uncertainty of this method has also highlighted: (i) the importance of improving the model used to reduce its structural error, and (ii) that suitable certified matrix reference material is lacking for isotopologue analysis [90]. Finally, ^13^C-MFA is used to determine flux values, but provides no mechanistic explanation for observed events. 

### 3.2. Label-Free Metabolomics Data Integration

Concentration data at a single point of time can be used to constrain a model by defining steady state concentrations, thermodynamics and, accordingly, reaction directionality or flux cue changes for multiple condition experiments.

To integrate single-point intracellular metabolomics into a kinetic model, one can utilize the steady state assumption, where the model is simulated until the system reaches a steady state that should describe the observed concentrations. Kinetic modelling can integrate metabolite concentration values, metabolic fluxes, enzyme amounts and/or kinetic parameters (i.e., Km or Vmax). Metabolomics experiments provide metabolite concentrations that can be used to either find the kinetic parameters given the known fluxes, or to infer the metabolic fluxes given the kinetic parameters known from the literature.

For instance, the kinetic parameter inference approach (find kinetic parameters from experimental data) was used to build personalized kinetic models of red blood cells based on patients’ metabolomics data [91]. Assuming that possible flux distributions in red blood cells of healthy individuals are significantly constrained, the authors calculated the parameters for 300 models, which constituted the model ensemble. All models in the ensemble converged on the same steady state flux with the assumption that possible distributions of fluxes in RBC of healthy individuals are significantly constrained. They showed that inferred parameters represented the genotype better than metabolite concentrations. The authors used this model to simulate a response to ribavirin, a drug often used in the treatment of hepatitis C, which is known to cause anemia. Since some of the individuals who participated in the study had protective SNPs, the authors were able to propose a mechanism for ribavirin-induced cell hemolysis.

One of the main challenges in the interpretation of metabolomics data is to trace back the changes that led to the observed phenotype. To address this issue directly with modelling, Savoglidis et al. [92] proposed a new methodology called inverse metabolic control analysis (IMCA), which was inspired by metabolic control analysis (MCA). MCA coefficients show the linearization of the response to the enzyme perturbation, i.e., they show whether and in what direction metabolite concentrations and reaction fluxes will change after the enzyme concentration is perturbed. IMCA on the other hand allows us to integrate metabolomics data and assess which enzyme concentration changes led to the measured metabolite concentrations. In this study, the authors developed a detailed kinetic model for sphingolipid metabolism and used it to integrate lipidomics data. They used the ORACLE framework [77] to generate a model ensemble, i.e., a cohort of models exploring possible solutions allowed within the range of the given kinetic and physico-chemical parameters, using computationally estimated flux and experimentally derived metabolite concentration measurements. They performed metabolomics on different strains of *S. cerevisiae* with knockouts in the sphingolipid biosynthesis. They used this data to validate their approach to predict knockout targets. The authors propose using this method for cell factory design and characterization of unknown mutations in personalized medicine.

Integrating metabolomics data from single-point experiments is challenging since constraint-based models only utilize flux information. The approaches mentioned so far require either strong assumptions or measurements of at least two time points to infer the metabolic fluxes. Nevertheless, different approaches have been proposed to incorporate single measurements. For instance, Mo et al. [93] used exometabolomics to formulate secretion constraints by introducing additional assumptions, e.g., a change in extracellular metabolite concentrations implies a change in exchange reaction flux. In this way, the authors were able to derive changes in the flux distribution. They studied different mutant strains of *Sacaromyces cerevisiae* in various environmental conditions and were able to characterize the changes in different parts of the metabolism.

Methods assessing the thermodynamics of the system also allow the integration of single-point absolute metabolite concentrations into models. Metabolite concentrations can be used to calculate the Gibbs energy of the reaction according to the equation:ΔrG = ΔrG° + RTlnQ,(3)
where ΔrG—Gibbs energy, ΔrG°—standard-state Gibbs energy, R—universal gas constant, T—temperature, Q—reaction quotient or $\frac{C^{c}D^{d}}{A^{a}B^{b}}$ where A and B are the concentrations of the substrates of the reaction, C and D are the concentrations of the products and a, b, c, d are respective stoichiometric coefficients. Thus, one can impose additional constraints on the system, since all reactions have to occur in the direction of decreasing Gibbs energy. Measuring the standard Gibbs energy change for all reactions taking place in the metabolism is challenging and not always possible. With the advent of the group contribution methods [94] and the ability to compute Gibbs free energies for any metabolic reaction [94,95], this method can be applied on large-scale models. This approach can be used as a data quality control by verifying that, for a given reaction, the calculated Gibbs free energy and the expected reaction’s flux are in agreement [96]. Additionally, the directionality of reversible reactions and the reaction Gibbs energy can be used to find enzymes that are likely to be allosterically regulated, since these reactions are generally thought to be far from equilibrium. For example, McCloskey et al. [97] used an *E. coli* genome-scale model as an integration platform to calculate the Gibbs free energy of the metabolic reactions in different *E. coli* strains. They showed significant differences in reaction thermodynamics, among them such drastic changes that led to reversing the reaction directionality. That enabled a more accurate characterization of the strains, making them good candidates for biotechnological applications. In another paper [98], researchers studied the metabolism of *E. coli* in anaerobic and aerobic conditions. They integrated metabolomics data as a way of estimating thermodynamics in a genome-scale model, which allowed them to find inconsistencies between the model and the data. With this analysis, the authors were able to highlight the importance of enzyme promiscuity and the usage of alternative pathways and pathway directionality for microbial growth. In a study using elementary flux modes, Gerstl et al. [99] introduced thermodynamic EFMA (tEFMA) which allows for the integration of metabolomic data with elementary flux modes. This information is used during the calculation of EFMs to exclude thermodynamically infeasible solutions and obtain biologically relevant EFMs, which allows for the evaluation of larger networks.

### 3.3. Integration of Metabolomics Data from Time-Series Experiments

A larger number of approaches have been developed to integrate metabolomics from time-course experiments compared to single-point metabolomics. In this section, examples of both intracellular and extracellular metabolomics time-series data integration will be discussed.

Exometabolomics from time-series experiments is one of the most commonly experimental data type integrated into both kinetic and constraint-based models. Exometabolomics refers to the metabolomics of the cell milieu, such as the culture media, the natural environment of the microorganisms, or the extracellular fluid. From an experimental point of view, this approach has less technical requirements than the assessment of intracellular metabolites. In addition, extracellular metabolite concentrations are integrated over time and produce a more intensive signal [100]. Such information allows us to define the boundaries of the system, in terms of maximum and minimum allowable exchange fluxes. Moreover, performing exometabolomics in a time-series manner opens a wide range of approaches for data integration. Time-series exometabolomic data allows us to infer secretion and uptake fluxes from the difference in concentrations and cell density over time. By using stoichiometric MFA (stMFA), the calculated fluxes can be integrated in the stoichiometric model to infer the resulting intracellular flux values. To illustrate, Quek et al. [101] used stMFA to determine the flux distribution in the central carbon metabolism of mammalian cells. From the extracellular concentrations of sugars and amino acids, Quek and co-workers inferred the exchange fluxes (i.e., uptake and secretion rates) and used them to fit the reaction fluxes in the mammalian central carbon metabolism model. Thus, the researchers were able to infer the intracellular flux distribution in hybridoma and CHO cell lines from uptake and secretion rates.

When there are distinct metabolic states throughout the time-series experiment (e.g., exponential and steady state phases during cultivation, behavior of the system before and after stress introduction, the adaptation to the stress), it is necessary to split the data into distinct phases and construct models separately to study the different metabolic behaviors in each phase [102]. Dynamic MFA (dMFA) was proposed as an alternative to such division, since it allows using a single model that describes the different metabolic states observed throughout the experiment. For example, Martinez et al. applied dynamic MFA to study the temperature shift in a CHO cell line fermentation study [103]. Changes in metabolite concentrations were approximated with B-splines, which were then used to calculate the fluxes continuously. It was shown that a temperature shift during cultivation leads to different time profiles of specific antibody productivity, with associated metabolic changes.

However, label-free metabolomics-based MFA has limitations. For instance, it cannot resolve the source of NADPH production in the cell. Hence, it requires us to make additional assumptions, such as cofactor balances or leaving out reactions that are believed to carry no flux. Still, this approach narrows down the number of possible solutions significantly and is relatively easy to apply.

An alternative approach is to use calculated fluxes to constrain the model for FBA. FBA will also use the assumption of metabolic optimality and give a single solution (or narrowed down solution for FVA). Aurich and coworkers [104] used this approach to integrate exometabolomics data from the NCI-60 panel (exometabolomics time-series data for 60 cell lines in duplicates [100]) to better understand cancer metabolism [105]. These data were used by the authors to calculate secretion and uptake rates (i.e., exchange fluxes), which were used to constrain the system’s boundaries. Here, they had to deal with a very common problem: targeted metabolomics is limited to a defined set of metabolites, thus not all exchange fluxes can be calculated. To address this issue, they proposed and utilized a new approach to calculate the missing uptake and secretion rates. The authors achieved it by finding the minimal set of additional metabolites that satisfied the growth rate, yet sufficiently constrained the model. The authors were thus able to describe distinct metabolic phenotypes within the cell lines. In addition, they performed perturbation analysis (gene knockouts) to predict novel drug targets and showed that the predicted flux values were in agreement with gene-expression and genomics data [104].

Dynamic FBA was proposed to incorporate the rate of change of flux constraints from measured concentrations in a time-course experiment [46]. Similarly to dMFA, dFBA enables the identification of changes in the flux distribution throughout the time of cultivation. Unlike stMFA methods, dFBA requires objective function in the form of a cellular objective rather than minimization of error between experimental and modelled data. Researchers studied the growth of *E. coli* on glucose in a batch fermentation, where acetate is initially secreted and subsequently consumed. With the use of dFBA, researchers were able to predict fluxes during diauxic growth on glucose and acetate by imposing additional constraints on the maximum oxygen uptake level, including the secretion and later reutilization of acetate [46].

M-DFBA and R-DFBA incorporate the MOMA and ROOM algorithms into dFBA. MOMA and ROOM were proposed to study the fluxes of knockout phenotypes. M-DFBA and R-DFBA were thus used to smooth flux transitions over time. Kleessen and Nikoloski [106] showed a good agreement between R-DFBA and a kinetic model simulation. Luo et al. [107] used the M-DFBA approach to study the metabolism of mammalian myocardia under normal and ischemic conditions. The authors proposed that, under ischemic conditions, myocardia does not maximize ATP production—an objective function often used for mammalian cell lines.

To incorporate time-series of both intracellular and extracellular metabolomics data, a number of dynamic FBA variations have been proposed. Among them are MetDFBA [108], TREM-Flux [109], and unsteady state flux balance analysis (uFBA) [110]. While these methods differ in implementation details and assumptions, they allow similar data integration processes. Time-series metabolomics and exometabolomics can be integrated in order to estimate the flux distributions. Using time-series metabolomics and exometabolomics, Bordbard et al. [110] were able to predict the fate of citrate in red blood cells, which had previously been mostly neglected. This finding was also validated by isotopic labelling. Chandrasekaran and colleagues [111] applied a similar approach to integrate time-series intracellular and extracellular metabolomics of naïve and primed murine pluripotent stem cells. They derived metabolomics data of mouse embryonic stem cells in naïve and primed states at three data points. The metabolomics data did not show any changes specific to a particular pathway, even though it showed significant changes in some metabolites. To address this, the authors developed a dynamic genome-scale metabolic model of the naïve and primed state. The approach, called FBA with flux activity coefficients, allows deviations from a strict steady state assumption [111]. The authors used time-course metabolomics measurements as cues for altered flux activity around a metabolite to find the state that best fits the metabolomics data. The authors performed both in silico knockout analysis and flux variability analysis to determine the range of feasible fluxes for each reaction. They showed that one-carbon metabolism differs between the two states, resulting in differential sensitivity to antifolates. They also found that Lin28 knockout cells, which were thought to change mouse embryonic stem cells to a naïve-like state, differ in the pentose phosphate pathway, suggesting that the states are not identical.

Kinetic models, which describe the changes of metabolite concentrations over time, are innately appropriate to integrate time-course metabolomics. As an example, Christodoulou et al. performed a time-series intracellular metabolomics experiment to study the metabolic response of *E. coli* to a peroxide pulse [112]. To further characterize the system, they built multiple kinetic models to test for important allosteric regulators that could lead to such response. The authors tested thousands of combinations of allosteric regulators and selected those that best described the metabolite concentration changes after the peroxide challenge. They found a number of relevant interactions that were further validated. Based on these findings, the authors proposed that *E. coli* reserves the pentose phosphate pathway flux capacity during normal growth in order to perform a rapid response for oxidative stress.

Another example is the work of Nishino et al. [113], where the authors tested different perturbations on a kinetic model in order to match the time-series metabolomics of long-stored red blood cells for transfusion. Red blood cells are stored at a low non-physiological temperature 4 °C. Since kinetic constants depend on the temperature, the authors had to find new parameters for their system rather than use systems that can be obtained from the literature data. They used a genetic algorithm to find the parameters that closely described the experimental data. Indeed, they showed that for some enzymes, the activity was altered in stored conditions compared to physiological conditions. The authors also performed simulations to describe the effects of possible additives on red blood cell metabolism, which were in agreement with known effects of blood storage.

### 3.4. Integration of Metabolomics with Multiple Omics Data

Integrating different types of omics data is the most promising, yet challenging, goal. It has the potential to incorporate enzyme regulation occurring at different levels, such as gene regulation or signaling pathways, and show the cross-talk between them. In this section, some of the most successful methods developed in the field will be discussed.

Hackett et al. collected omics data (proteomics, fluxomics, metabolomics) for 25 different steady state conditions for *S. cerevisiae* grown in a chemostat [114]. They developed a method (SIMMER) that tests whether the observed fluxes through an individual reaction can be explained by a Michaelis–Menten rate law, given the metabolite and enzyme concentrations measured in the experiment. When this is not the case, the algorithm tries to find the allosteric regulator that best explains the measured flux. Interestingly, this approach takes into account one reaction at a time. While it is possible that this leads to emergent properties being neglected, it is compatible with the analysis of large-scale models. It was shown that including allosteric regulators improved the fit for many reactions, while several known interactions were recovered and new regulations were proposed. Researchers also used their model to study the governing principles of metabolic regulation of *S. cerevisiae*. They showed that regulation occurs not only within pathways but also between pathways.

The same dataset from Hackett and colleagues was also analyzed by St John et al. [69]. In this study, the authors modeled the whole system simultaneously using the lin-log formalism to model the reactions dynamics instead of Michaelis–Menten kinetics. This method allows for kinetic parameter inference and selection of allosteric regulators. The authors were able to recapture most of the variance seen in observed fluxes, enzyme and metabolite concentrations. Additionally, by using a Bayesian approach, the priors for the parameters can be specified, thus encoding additional beliefs about the system. Such methods have been used for a long time. However, recently they are becoming more relevant with the rise of probabilistic languages (e.g., PyMC3 [115], stan [116], Turing [117]) and the development of the computation capabilities needed for such analyses.

Untangling complex interactions between signaling and metabolism is one of the most important goals of metabolic modelling. Such a possibility was demonstrated by Yugi and coworkers [118]. In their study, they focus on the dynamic events during insulin activation of the rat hepatoma Fao cell culture. They obtained time-course phospho-proteomics and metabolomics and built a kinetic model to explain the dynamic changes seen in experiments. The model included both the metabolism and signaling pathways. A possible flux distribution was sampled from the stoichiometric model, and used to estimate the kinetic model parameters. The kinetic model was then used to explore the behavior around the liver-type phosphofructokinase 1 node. Experimental values for metabolite concentrations were used in that model and kept at a fixed level, except for fructose-1,6-bisphosphate’s concentration. The authors describe how they iteratively refined the kinetic model to capture the specific differences observed in fructose-1,6-bisphosphate, thus showing that insulin regulates the glycolytic pathway and that the key event is the phosphorylation of phosphofructokinase.

A number of approaches aim to estimate the flux distribution without using labelled data. To address that, Yizhak et al. [119] proposed a method that finds the steady state flux distribution by integrating metabolomics and proteomics data into a genome-scale model. Researchers integrated metabolomics and proteomics together in kinetic modelling simulation of a core set of reactions and used predicted fluxes through those reactions to constrain genome-scale model. In addition, the predicted fluxes have to be consistent with flux estimations derived from kinetic model simulations, which integrate metabolomics and proteomics data. They demonstrated that their approach is able to predict the metabolic state of human erythrocytes and *E. coli* with different gene knockouts. Kleessen et al. [109] proposed the approach TREM-Flux to integrate time-series transcriptomics and quantitative metabolomics data to predict the intracellular flux distribution. TREM-Flux uses the E-Flux approach [120] in which the reaction flux is assumed to be proportional to the expression value of the gene whose product catalyzes the reaction, and then estimates flux between each time point from the changes of the metabolites level. The authors used this method to characterize the systemic metabolic response of a *Chlamydomonas* strain to rapamycin treatment. Covert et al. [121] also proposed an integrated FBA (iFBA) framework that combines dynamic and constraint-based models to address inclusion of the regulatory behavior in constraint-based models. Their approach combines FBA with regulatory Boolean logic and ordinary differential equations. Thus, researchers were able to show the possibility to add regulatory (both expression regulation and signal transduction) components both to FBA and dynamic models.

Metabolomics and transcriptomics data can also be used to impose constraints on constraint-based genome-scale metabolic models. GIM3E is an example of such an algorithm [122]. The GIM3E algorithm uses metabolomics, both absolute and relative, and transcriptomics data to develop constraints for a constraint-based model. Specifically, this algorithm uses metabolomics data as cues to ensure that the detected species are used in the model and transcriptomics data to further constrain fluxes proportionally to the transcript level in different conditions. Notably, metabolomics and transcriptomics imposed different constraints on the solution space. This approach is practical in large-scale models and explorative studies. 

Salvy and Hatzimanikatis [123] published a variation on metabolism and expression models, also referred to as ME models [124], named Expression and Thermodynamics Flux models (ETFL), which are capable of incorporating metabolomics data as thermodynamic constraints that ensure proper directionality of reactions. In this approach, thermodynamically feasible fluxes are combined with proteomics and transcriptomics data. ETFL can predict proteome-limited growth by integrating growth-dependent parameters (e.g., relative protein or mRNA concentrations) and phenotypic differences in growth regimens. Additionally, the method was able to predict gene-editing experiments more accurately than the FBA method.

## 4. Multicellular, Multitissue and Community Modelling

The modelling approaches described so far treat a whole cell population as a homogeneous single-cell type group of cells. While this is a simple yet powerful approach that allows obtaining numerous findings, it is challenging to study systems where such assumptions do not hold. Examples of such systems are (a) interactions within communities of microorganisms in different environments, (b) cross-talk between cells in a normal tissue, (c) heterogeneous tumors, (d) interactions between tissues and organs, (e) interactions within the microbiome, (f) interactions between the microbiome and an organism, etc. In order to study these biological systems, novel approaches capable of capturing the underlying interaction between different cell types are required. The interest in these topics is driving the rise in the development of multicellular, multitissue, and community modelling techniques and software. The necessity to work with different cell types increases the size and complexity of the models. Strategies for building community metabolic models are usually the following (1) multispecies dynamic modelling that accounts for kinetics of nutrient uptake and metabolite production by individual species; (2) compartmentalized network modelling, where species are treated as compartments in the model, and generally used for prediction of cross-species metabolic interactions; (3) mixed-bag network modelling, where the community is treated as a superorganism (cell). These strategies are typically used to uncover environment-community interactions [125]. Yet, currently available experimental approaches are not able to capture many of these interactions. All this leads to an increase in the degrees of freedom, which increases the uncertainty of metabolic model solutions. Multilevel modelling is, however, crucial to describe which interactions occur between different cell types and how they affect the individual cell functioning and the whole community behavior. Metabolic modelling has been utilized to decipher interactions and mechanisms in mutualistic and parasitic relations [126]. FBA techniques and network analysis approaches are already being used to study the communities [127,128,129]. For example, metabolic modelling approaches have been applied to study how diet affects microbiome composition [130,131]. In these studies, the researchers were able to predict changes in metabolite concentrations induced by microbiome composition [131].

### 4.1. Metabolomics Data Integration for Community Models

Mature metabolic modelling standards and practices are the foundation for successful data integration processes [132]. Researchers are already applying metabolic modelling to integrate genomics and transcriptomics data. A number of approaches have already demonstrated that integration of metabolite concentration data into mechanistic models of multicellular systems is viable.

Dynamic constraint-based methods are suitable for integration of large-scale models, since they do not rely on kinetic parameters that are often hard to obtain. dFBA was one of the first methods to be used in metabolomics data integration for community modelling. Hanemaaijer and coworkers applied dFBA to study a synthetic anaerobic co-culture of Clostridium acetobutylicum and Wolinella succinogenes [133], and showed that dFBA can be used to study community interaction. These bacteria interact by exchanging hydrogen with each other. The changes in hydrogen concentration were used to predict the intracellular fluxes and optimal metabolite exchange rates. 

Inference of fluxes is crucial for the understanding of metabolism and its regulation. In addition, it facilitates the development of kinetic models that allow researchers to characterize the system and its regulation. ^13^C-MFA is a gold standard to infer flux distributions and has been extensively applied to mono-cultures during the past two decades. Yet, there are only a handful of ^13^C-MFA examples for microorganism communities. Ghosh et al. demonstrated a peptide-based alternative to the standard amino acid-based ^13^C-MFA [134]. This method is based on the availability of peptide sequences that can be used to trace back the microbial species they originate from and used to infer intracellular metabolic fluxes. Recently, Gebreselassie and Antoniewicz [135] combined total biomass labelling experimental data and ^13^C-MFA to infer the metabolic flux profile of a co-culture system of two *E. coli* knockout strains. They did not use physical separation nor genetic labelling to perform the measurements. Rather, the fluxes were inferred via pooled distribution of isotopes in co-culture. While the peptide-based approach is capable of resolving fluxes in larger and more diverse communities, the total biomass labelling approach is suitable for closely related species, and in theory should be capable of finer resolutions, since peptides inherently contain less flux information than amino acids.

### 4.2. Metabolomics Data Integration for Multicellular, Multitissue and Multiorgan Models

Constraint-based modelling has been used to model whole organisms, multitissue and multicellular models of plants and animals to elucidate the interactions between the different cell types within a tissue and higher levels of interactions [136,137,138]. To integrate the changes in metabolite concentrations, dFBA has been applied to whole-plant metabolism, including diurnal changes [139,140], and to a human whole-body physiologically-based pharmacokinetic model [141]. For example, Krauss et al. applied dFBA to describe the human metabolism within the context of whole-body physiologically-based pharmacokinetics [141]. They demonstrated three cases in which this approach was successful: (1) the analysis of the distribution and therapeutic effect of allopurinol in the treatment of hyperuricemia; (2) the effect of impaired ammonia metabolism on blood plasma metabolites levels to identify biomarkers; (3) paracetamol-induced intoxication on liver metabolism.

Recently, ^13^C-MFA was applied at an organism level in humans [142]. The authors explored the contribution of circulating lactate and related metabolites to tumors metabolism. Patients were infused with labelled glucose and both isotope enrichment analysis [142] and MFA were performed to analyze the lactate metabolism. They were able to provide the first direct evidence that lactate and other three-carbon products of glycolysis are consumed by human non-small-cell lung cancers. 

## 5. Conclusions and Perspectives

The technical capability provided by different high-throughput platforms to analyze and quantify an increasing number of molecular entities existing within a living cell opens up opportunities for novel discoveries and scientific findings in metabolic research [91,143,144,145]. To cope with this vast amount of omics data, systems biology has developed mathematical approaches to understand emergent properties of metabolism. In this sense, metabolic modelling can be applied to different metabolic studies and has the capability to encode previous knowledge to extract more information from the data or find existing inconsistencies in current knowledge and propose new hypotheses and new experimental designs in an iterative fashion. For instance, mechanistic models are a promising tool for drug design and personalized medicine [91,143], and are capable of predicting response to drug combination based on single drug response data [146]. There are examples of how metabolic engineering tasks are guided by metabolic modeling for increased production of industrially relevant compounds [144,145]. Broadly, the application of mechanistic models was very fruitful in metabolism research, with high potential in the area of data integration.

Stoichiometric models have the advantage of scaling well for large-scale problems, since they require little prior information. In this review, we have discussed several techniques and examples to integrate metabolomics data in order to study cell metabolisms. In general, kinetic models are more suitable for well-known metabolic systems, which on the other hand reduce their scope to models of small and medium size. Kinetic models are also suitable to predict the state of the systems and they can deal with allostery and regulation. On the other hand, constraint-based modelling is usually more suitable when using large-scale models with scarce knowledge about the system. We refer to the specialized article for the details of usage of mechanistic models [147]. The type of data and the system under study also play an important role when choosing integration techniques. For example, dynamic techniques (i.e., D-FBA) should be used when the overall metabolic behavior of a system is researched that is not constant along the experiment. Another example is the use of labelling data when there is need of a more detailed description of intracellular fluxes. We have also seen that gathering time-series data can dramatically improve the possibilities in modelling and interpreting the data.

The application of such models supports the systematic overview of the metabolism by applying physico-chemical principles to metabolic pathways. Kinetic models can innately encode different types of information and various types of omics data from first principles. Kinetic models, especially large scale, are not easy to parametrize due to the lack of kinetic parameters describing the in vivo behavior or because they are not fully characterized and not applicable for case-specific studies. Accurate inference of large-scale kinetic models from experimental data is the most promising technique, yet needs a significant amount of data points and efficient computational approaches as well as computational power. The combined application of both kinetic models and constraint-based models in the study of biological systems would allow taking advantage of their complementary features in a synergistic manner with the potential to overcome existing setbacks. The combination of the methods can be achieved in different ways, from modelling different parts of the metabolism with either kinetic modelling or constraint-based modelling [119,121] (or other formalisms) to enhancement of constraint-based models with kinetic parameter information, such as in the GECKO method [7]. Another hybrid approach is to combine mechanistic and statistical or machine learning models. It is challenging to identify the causalities and mechanisms by using machine learning approaches. Recently, the article showing “white-box” approach [148], as opposed to “black-box” uninterpretable machine learning models, was shown, demonstrating the power of such methods. Modern machine learning algorithms are capable of pinpointing associations from dataset inputs and outputs but are not able to give mechanistic explanations of the phenomena. To address this challenge, researchers used an integrated screening–modeling–learning approach in which they linked input and output data with mechanistic models. White-box machine learning is potentially capable of identifying biologically interpretable biomarkers, personalized treatment strategies and mechanisms of disease pathogenesis from big datasets.

The development of metabolome measurement technology also plays an important role in the accuracy and comprehensiveness of metabolic modeling. The most important directions are accurate quantitative acquisition, identification of unknown metabolites and subcellular metabolite localization. First, acquisition of quantitative metabolome data can contribute to improvement of model accuracy. Therefore, progress in a comprehensive metabolite measurement method using the stable isotope dilution method is desired [149]. Furthermore, identification of unknown metabolites is also an important issue in constructing an accurate metabolic model [150]. It can uncover shed lights on new reactions and better determine scope of the model. Metabolomics data integration dealing with different cell compartments remains a challenge. Existing experimental methods usually capture pooled metabolite concentrations. For example, for eukaryotic cells, it is not possible to obtain the real concentration and/or isotopic labelling in different compartments from the pooled data. Both experimental and computational improvements are needed to solve this issue. So far, different experimental approaches have been proposed for sample preparation to extract mitochondria and leave the internal metabolites intact [151,152,153,154]. However, only mitochondria have been isolated for a labeled and label-free analysis of the mitochondrial metabolites. Therefore, studies are limited where researchers are focusing on metabolites that only exist in cytosol and/or mitochondria, while the metabolism in other cell compartments is neglected.

Multicellular modelling of heterogeneous cellular ecosystems is now an exciting and quickly developing field. Researchers are still discovering new mechanisms of interplay between organs and tissues in different whole-organism studies. Awareness of the importance of tumor heterogeneity in clinical outcome has led to more studies on the metabolic vulnerabilities of tumor subpopulations [155]. Following the success of metagenomics and metatranscriptomics, studying the metabolic basis of microbiome and organism interaction will demand systems biology approaches to integrate, interpret, and generate new hypotheses. First efforts show that widely used methods currently applied to single-cell type modelling can be extended to multicellular approaches describing metabolic interactions between different cell types [156]. The recent advances in single-cell metabolomics [156] have the potential to support improvements in methods to analyze heterogenous cellular communities. However, more experimental multicellular-specific methods are also needed.

Standardization of metabolic model formulation and simulation is an important task for researchers and the scientific community in order to advance the field of metabolic modelling and data integration in particular. Several initiatives and tools were proposed in order to harmonize the modelling studies, including MEMOTE (for metabolic model tests) [157] and The Systems Biology Markup Language (SBML) [158]. Further, good practices of models and data distribution should be more widely adopted in order to allow different scientific groups to integrate different datasets and use already published models. We refer the readers to the Findability, Accessibility, Interoperability, and Reusability (FAIR) guidelines [159].

Systems biology already provides insights into the mechanisms that govern metabolism in different biological fields. By routinely using mechanistic models to integrate and interpret metabolomics data, it will be possible to shed more light into the complex network of interactions in the metabolism.

## Figures and Tables

**Figure 1 metabolites-10-00303-f001:**
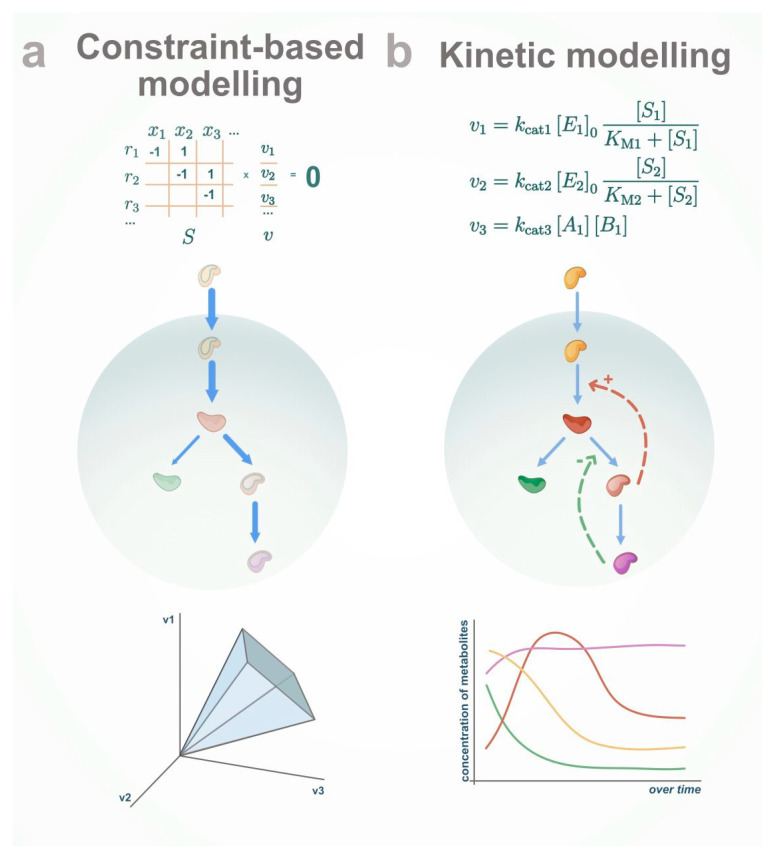
Modelling approaches to study metabolism. (**a**) Constraint-based modelling allows balancing the fluxes in the system, but cannot work with metabolite concentration directly. (**b**) Kinetic modelling allows the simulation and analysis of the dynamic behavior of metabolite concentration over time.

**Figure 2 metabolites-10-00303-f002:**
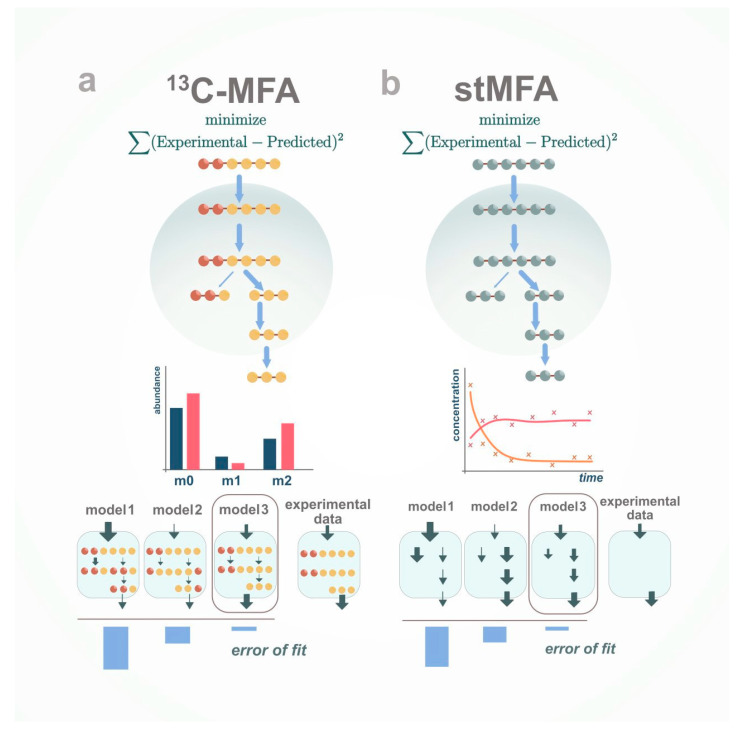
Metabolic flux analysis. MFA is an optimization problem that minimizes the difference between simulated and experimental flux data and labelling pattern data in case ^13^C-MFA and solely flux data in case stMFA. (**a**) ^13^C-MFA relies on balancing measured and unmeasured rates and patterns of isotopic labelling given the metabolic model; (**b**) stMFA relies solely on balancing measured and unmeasured rates given metabolic model.

**Figure 3 metabolites-10-00303-f003:**
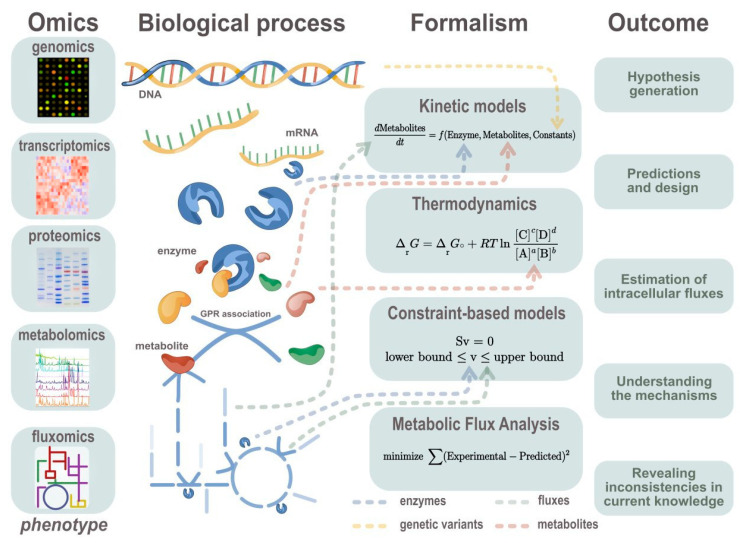
Description of reviewed approaches to integrate omics data and the outcome of the modelling methods. Different omics data types can be generated in order to study different metabolism phenomena. Those omics data correspond to different layers of cell functioning. Collected omics data can be formalized in different modelling approaches as different layers of the hierarchical organization of biological systems. Conventional ways of integration are shown with arrows pointing to the equation part. The typical outcomes of the modelling approaches described in this review are highlighted.

**Table 1 metabolites-10-00303-t001:** Summary of the metabolomics integration approaches described in the review.

Integrated Omics Data	Model Type (FBA, MFA, Kinetic Model, etc.)	Comment	Reference
Isotopic-labelling data	^13^C-MFA	^13^C-MFA at a genome scale	[88]
	^13^C-MFA	^13^C-MFA of central carbon metabolism of hepatocellular carcinoma and effect of Hexokinase-2 on the metabolism	[82]
	^13^C-MFA	^13^C-MFA at a genome scale	[53]
	^13^C-MFA	^13^C-MFA at a genome scale of evolved knockout *E. coli* strains	[89]
	^13^C-MFA	^13^C-MFA of central carbon and amino acid metabolism reveals how changing medium amino acid composition metabolism in CHO cell culture	[83]
	^13^C-MFA	Application of ^13^C-MFA of central carbon and amino acid metabolism to study CHO cells with high productivity of industrially relevant proteins	[84]
Metabolomics (single data point)	Kinetic model	Personalized kinetic model parametrization and analysis of red blood cells	[91]
	Kinetic model	IMCA approach to trace back the changes that led to the observed phenotype	[92]
	Constraint-based model	Constraint-based modelling approach and single-point extracellular metabolomics	[93]
	Constraint-based model	Tool for system thermodynamic analysis of quantitative metabolomics	[96]
	Thermodynamic FVA	Genome-scale thermodynamic FVA applied to integrate the metabolomics data of different industrial strains of *E. coli*	[97]
	Thermodynamic EFM	Combination of thermodynamic and EFM analysis	[99]
	Constraint-based model	Genome-scale thermodynamic CBM applied to integrate metabolomics of *E. coli* to research aerobic and anaerobic metabolism	[98]
Time-series metabolomics data integration	Stoichiometric MFA	stMFA of carbon central metabolism in mammalian cell culture	[101]
	Dynamic stoichiometric MFA	Dynamic stMFA of carbon central metabolism used to study the effect of the temperature shift on CHO	[103]
	FBA	Genome-scale FBA for cancer cell line metabolism analysis	[104]
	dFBA	dFBA at a genome scale used to study diauxic growth in *E. coli*	[46]
	MetDFBA	dFBA variation used to integrate time-series metabolomics data	[108]
	uFBA	dFBA variation used to integrate time-series metabolomics data and study the metabolism of red blood cells	[110]
	M-DFBA	dFBA variation to integrate time-series metabolomics data to study myocardial metabolism under normal and ischemic conditions	[107]
	R-DFBA	dFBA variation to integrate time-series metabolomics data	[106]
	FBA with flux activity coefficients	FBA with time-course metabolomics measurement cues for altered flux activity around a metabolite to study the metabolism of pluripotent stem cells	[111]
	Kinetic model (Michaelis–Menten laws). Parameters known (sampled across the literature values to account for uncertainty)	Kinetic models used to find key regulations in the metabolism to study the response of metabolism on oxidative stress in *E. coli*	[112]
	Kinetic model	Kinetic model of central carbon metabolism of long-stored red blood cells to describe the metabolism changes at not-standard temperature	[113]
Multiomics data integration	Kinetic model	Kinetic model used to find new regulators	[114]
	Kinetic model	Kinetic model used to find new regulators	[69]
	Kinetic model	Integration of metabolomics and phosphoproteomics into a kinetic model to characterize the response to the insulin on the signaling and metabolic level	[118]
	Kinetic model and constraint-based model	Flux estimation from single-point unlabeled data by integrating it into a model which consists of a kinetic and constraint-based model	[119]
	FBA with regulatory Boolean logic and kinetic model	Genome-scale FBA modification that captures metabolism, regulation and signaling *in E. coli*	[121]
	FBA and other constraint-based methods	GIM3E, an approach to develop condition-specific models	[122]
	ME model	Thermodynamically consistent ME model	[123]
	TREM-Flux	dFBA variation used to integrate time-series metabolomics and transcriptomics data to study the response of *Chlamydomonas* to rapamycin treatment	[109]
Multicellular, multitissue and community modelling	DFBA	dFBA for co-cultures to study the metabolic interactions in microbial community	[133]
	^13^C-MFA	A peptide-based ^13^C-MFA approach for co-cultures	[134]
	^13^C-MFA	^13^C-MFA for co-cultures	[135]
	FBA	FBA approach to study plant on the organismal level, highlighting differences in tissue-specific metabolic networks	[136]
	FBA	Cell-specific metabolic models are combined within single model allowing the study of complex physiological processes such as a Cori or alanine cycle	[137]
	FBA	A modelling study accounting for the interactions between cell types found in the brain is validated with experimental data and demonstrates metabolic interplays and activities that less detailed models are missing	[138]
	FBA + FPM (functional plant model, special kind of kinetic model)	Integration of dynamic FPM and static FBA allowed for a whole-plant time-resolved analysis	[139]
	dFBA	Combination of dFBA with resource allocation prediction applied to the whole-plant model	[140]
	dFBA within PBPK	Application of multiscale modelling to hepatocyte metabolism and physiology	[141]
	^13^C-MFA	^13^C-MFA on whole-body level to trace the fate of lactate in human lung tumors	[142]

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
