# Peer review of "Metabolic Modelling as a Framework for Metabolomics Data Integration and Analysis"

_metabolites, 2020, doi:10.3390/metabo10080303_

Round 1

Reviewer 1 Report

This is generally an interesting review about computational methods used for metabolic modelling. It introduces constraint-based and kinetic modelling and describes recent developments and applications, especially MFA.

I have only few suggestions. From a computational point of view there are essentially two approaches: Steady-state (solving equation systems) and kinetic modelling which boils down to solving systems of differential equations. The latter is computationally much more expensive, and hence limited. But where it can be applied it is very powerful. I don't see any example cited. Especially MFA was traditionally done by kinetic modelling (all the work by Vitali and Cascante, also Zamboni and Sauer), but many others hav used the same. In a review like this I would like to see why kinetic modelling gets more and more abandoned.

Moreover, the review would greatly benefit from some extra structuring that shows the reader clearly where a form of contraint based modeling is used. I find this particularly confusing for MFA where the expression MFA already suggests that a flux is modelled. But in all examples show it is done by the steady-state approach.Maybe a list of the FBA type methods would help.
It is not celar what the difference between stMFA and dFBA is - further explanation would be nice.

What I also miss is a a more global critical view of these methods (there is some on page p, line 183). A lot has been predicted and modelled. What are these models needed for and what is their predictive power? If you can't model allostecric regulation you can't model glycolysis. Are there examples dealing with this?

What are the majour applications beyond filling pdf's in journals?I see 2-3 examples from drug discovery which are interesting. 

There are also some small issues:

l78 In the following?
l128, something wrong with thte formula
13C not superscripted, same for CO2 not subscripted
Reference 60 must be wrong
l481: label-free
l595: reactions to constrain (no t in the end) - this error occurs at least twice
l595: Besides, it has to be

Author Response

Response to comments made by Reviewer

-This is generally an interesting review about computational methods used for metabolic modelling. It introduces constraint-based and kinetic modelling and describes recent developments and applications, especially MFA.

We are pleased with the positive assessment of our work.

-Especially MFA was traditionally done by kinetic modelling (all the work by Vitali and Cascante, also Zamboni and Sauer), but many others have used the same.

We appreciate the reviewer comment. In the present manuscript, we highlight the work published by Cascante, Selivanov since it represents some of the most relevant papers in the field of KM applied to MFA. We refer to that approach in lines 294-300 “Notably, this approach (minimization of the difference between observed and modelled fluxes and isotopic pattern) can be applied in the kinetic modelling framework and such application was demonstrated [58–62]. In this approach, kinetic parameters are refined and the flux distribution is determined. This approach was not widely explored, despite clear benefits: acquiring not only flux distribution that describes the metabolic phenotype, but also a kinetic model that has predictive and explanatory capacities.”. Following the suggestion of the reviewer we added more references from Zamboni, Sauer, Nöh and Wiechert to that approach. We also describe the application of such methods in the metabolomics integration section in Lines 407-421 “Aguilar and colleagues studied metabolic reprogramming associated with cancer stem cells in a dual cell model consisting of one highly metastatic subpopulation displaying cancer stem cells properties and a second non-metastatic subpopulation displaying a stable epithelial-mesenchymal transition [57]. Experiments with incorporation of 13C-glucose were used to estimate the flux distribution and showed differences in cellular respiration between those two cell types. In addition, model-driven strategies were used to support other findings in a study that suggest differences between metabolic features of cancer stem cells andepithelial-mesenchymal transitionphenotypes. Each of the phenotypes can be linked to optimal cell proliferation and sustaining specific cell phenotypes. This study provides an example of how isotope-labelling data can be used with kinetic models to determine the intracellular fluxes. Labeling metabolomics in combination with kinetic modelling can also be used to infer the intracellular metabolic network topology in order to study physico-chemical phenomena like channeling [80]or crowding [81].”

-In a review like this I would like to see why kinetic modelling gets more and more abandoned.

Indeed kinetic modelling has its limitations. The researchers have to overcome many challenges due to lack of kinetic parameters and limitation in size. Currently there is a development in that field and many solutions already available. In the reviewed version of this manuscript we have included some further explanation to make the current state of the development more clear in Lines 335-343:

“Researchers are working on solutions to address current limitations, such as kinetic parameters paucity and small scale preference. One possible approach to address the lack of kinetic parameters and/or uncertainty in the given kinetic parameters, and as a consequence in predictions, is to use ensemble modelling [76]. Among currently available ensemble modelling packages are approaches like ORACLE [77], GRASP [78]. An update to the previous work is K-FIT [74]which significantly improves computational efficiency and makes it more accessible, enabling to extend the use of kinetic modeling to larger network systems.” In addition, there are a lot of progress towards hybrid methods between constraint-based modelling and kinetic modelling in order to combine strengths of both approaches. We refer to lines 885-891: ”The combined application of both kinetic models and constraint-based models in the study of biological systems would allow to take advantage of their complementary features in a synergistic manner with the potential to overcome existing setbacks. The combination of the methods can be achieved in different ways, from modelling different parts of the metabolism with either kinetic modelling or constraint-based modelling [119,121](or other formalisms) to enhancement of constraint-based models with kinetic parameters information, such as in GECKO approach [7]. ”

-Maybe a list of the FBA type methods would help.

There are numerous FBA approaches that were developed in order to address different scientific questions. In order to not go beyond the scope of the paper and provide the information about existing methods, we decided to refer the readers to exhaustive specialized reviews. References can be found in the text lines 208-215 “In this review we described those constrained-based methods, that are necessary to understand state-of-the-art approaches of metabolomics data integration. To get more information about the variability of approaches of constraint-based methods we refer the readers to the review by Lewis et. al [29]. We also refer to other articles covering other powerful and widely used techniques such as GECKO method [7], a method that enhances constraint-based methods to account for enzymes maximum capacity by incorporating kinetic constants, metabolism and expression models(ME-models) [30] and protein allocation models [31–34].”

-It is not clear what the difference between stMFA and dFBA is - further explanation would be nice.

Following the reviewer suggestion we have added a more detailed description of these methods to clarify the difference between them - lines 608-610 “Unlike stMFA methods, dFBA requires objective function in form of cellular objective rather than minimization of error between experimental and modelled data.” In addition, you can find the explanation in the text Lines 237-241 : “MFA follows the same principles as FBA but does not assume any functional objective, e.g. growth rate or ATP production. Instead, MFA is an optimization problem that minimizes the difference between simulated and experimental flux data for constraint based models”

- What I also miss is a more global critical view of these methods (there is some on page p, line 183). A lot has been predicted and modelled. What are these models needed for and what is their predictive power? If you can't model allostecric regulation you can't model glycolysis. Are there examples dealing with this?

Following the reviewers suggestion we added the text to pay particular attention to the methods limitation in the conclusion and perspectives section. See lines 860-876”Stoichiometric models have the advantage of scaling well for large-scale problems, since they require little prior information.In this review we have discussed several techniques and examples to integrate metabolomics data in order to study cell metabolisms. In general kinetic models are more suitable for well known metabolic systems, which on the other hand reduce their scope to model of small and medium size. Kinetic models are also suitable to predict the state of the systems and they can deal with allostery and regulation. On the other hand, constraint-based modelling is usually more suitable when using large-scale models with scarce knowledge about the system. We refer to the specialized article for the details of usage of mechanistic models [147]. The type of data and the system under study also plays an important role when choosing integration techniques. For example, dynamic techniques (i.e. D-FBA) should be used when the overall metabolic behavior of a system is researched, that is not constant along the experiment. Another example is the use of labelling data when there is need of a more detailed description of intracellular fluxes. We have also seen that gathering time-series data can dramatically improve the possibilities in modelling and interpreting the data.” and lines 877-888 “Application of such models supports the systematic overview of the metabolism by applying physico-chemical principles to metabolic pathways. Kinetic models can innately encode different types of information and various types of omics data from first principles. Kinetic models, especially large scale, are not easy to parametrize due to the lack of kinetic parameters describing the in vivo behavior or because they are not fully characterized and not applicable for case-specific studies. Accurate inference of large scale kinetic models from experimental data is the most promising technique, yet needs a significant amount of data points and efficient computational approaches as well as computational power. The combined application of both kinetic models and constraint-based models in the study of biological systems would allow to take advantage of their complementary features in a synergistic manner with the potential to overcome existing setbacks.” Also, we address the limitations and comparisons between methods throughout the Metabolic modelling approaches section. For example: Lines 93-104, 159-160, 187-189, 201-203, 224-233, 241-246, 309-312, 317-319, 323-327

We explain what kinetic models are needed for in lines 304-316: “Kinetic modelling describes the changes in metabolite concentrations given an initial condition by representing the metabolic network as a system of differential equations, most commonly ordinary differential equations. The fate of each metabolite is described by an ordinary differential equation that consists of a rate equation for each reaction that either consumes or produces the metabolite. Solutions for such systems yield the concentration of metabolites at each time point, and make it possible to evaluate the evolution of fluxes and metabolites over time. The equations for the reaction rates are functions of the enzyme concentrations, kinetic parameters, and metabolite concentrations. Kinetic modelling describes not only the stoichiometric relation between substrate and products but also the enzymatic mechanisms and different levels of regulation (i.e. allostery or post-transcriptional regulation), which makes these methods suitable for the integration of multiple omics data types.” In order to be more clear what constraint-based methods are used for we added the explanation to the text where we introduce constraint-based methods. Lines 132-133 “By solving this system of equations it is possible to get the flux distribution.” Also, the comparison of the methods and reference to the more detailed specialized article were introduced in the discussion and perspectives session. Lines 860-876 “Stoichiometric models have the advantage of scaling well for large-scale problems, since they require little prior information. In this review we have discussed several techniques and examples to integrate metabolomics data in order to study cell metabolisms. In general kinetic models are more suitable for well known metabolic systems, which on the other hand reduce their scope to model of small and medium size. Kinetic models are also suitable to predict the state of the systems and they can deal with allostery and regulation. On the other hand, constraint-based modelling is usually more suitable when using large-scale models with scarce knowledge about the system. We refer to the specialized article for the details of usage of mechanistic models[147]. The type of data and the system under study also plays an important role when choosing integration techniques. For example, dynamic techniques (i.e. D-FBA) should be used when the overall metabolic behavior of a system is researched, that is not constant along the experiment. Another example is the use of labelling data when there is need of a more detailed description of intracellular fluxes. We have also seen that gathering time-series data can dramatically improve the possibilities in modelling and interpreting the data.”

-If you can't model allostecric regulation you can't model glycolysis. Are there examples dealing with this?

In the following review we included work that was dedicated to finding the missing metabolic regulators in order to explain the phenotype. See Lines 648-659 “Kinetic models, which describe the changes of metabolite concentrations over time, are innately appropriate to integrate time-course metabolomics. As an example, Christodoulou et al performed a time-series intra-cellular metabolomics experiment to study the metabolic response of E. coli to a peroxide pulse [112]. To further characterize the system they built multiple kinetic models to test for important allosteric regulators that could lead to such response. The authors tested thousands of combinations of allosteric regulators, and selected those that best described the metabolite concentration changes after the peroxide challenge. They found a number of relevant interactions that were further validated. Based on these findings the authors proposed that E. coli reserves the pentose phosphate pathway flux capacity during normal growth in order to perform a rapid response for oxidative stress.” or lines 677-698 “Hackett et al collected omics data (proteomics, fluxomics, metabolomics) for 25 different steady-state conditions for S. cerevisiae grown in a chemostat[114]. They developed a method (SIMMER) that tests if the observed fluxes through an individual reaction can be explained by a Michaelis-Menten rate law, given the metabolite and enzyme concentrations measured in the experiment. When this is not the case the algorithm tries to find the allosteric regulator that best explains the measured flux. Interestingly, this approach takes into account one reaction at a time. While it is possible that this leads to emergent properties being neglected, it is compatible with the analysis of large-scale models. It was shown that including allosteric regulators improved the fit for many reactions, while several known interactions were recovered and new regulations were proposed. Researchers also used their model to study the governing principles of metabolic regulation of S. cerevisiae. They showed that regulation occurs not only within pathways but also between pathways.The same dataset from Hackett and colleagues was also analyzed by St John et al [69]. In this study, the authors modeled the whole system simultaneously using the lin-log formalism to model the reactions dynamics instead of Michaelis-Menten kinetics. This method allows for kinetic parameter inference and selection of allosteric regulators. The authors were able to recapture most of the variance seen in observed fluxes, enzyme and metabolite concentrations. Additionally, by using a Bayesian approach the priors for the parameters can be specified, thus encoding additional beliefs about the system. ” However, systematic assessment of predictive power of the models is beyond the scope of this review. We think provided case studies give enough of an overview to help readers to navigate the field and choose the methodology which is the best in the specific case.

-What are the major applications beyond filling pdf's in journals?I see 2-3 examples from drug discovery which are interesting.

Mechanistic models and metabolomics integration provide researchers valuable tools to investigate causality, make better predictions, guide further experiments by generating hypotheses, understand mechanisms and reveal inconsistencies in current knowledge. This review aimed to show different outputs of the modelling processes to give the reader a perspective on what it is possible to expect from addition modelling toolbox into their studies of metabolism. In order to give a more wider examples of successes of mechanistic modelling we have added some references to the Discussion and Perspectives section: Lines 853-859 “For instance, mechanistic models are a promising tool for drug design and personalized medicine [91,143], are capable of predicting response to drug combination based on single drug response data [146]. There are examples of how metabolic engineering tasks are guided by metabolic modeling for increased production of industrially relevant compounds [144,145]. Broadly, application of mechanistic models was very fruitful in metabolism research with high potential in the area of data integration.“

All minor comments are addressed.

Reviewer 2 Report

Volkava et al provide a very comprehensive review on metabolic modelling and the integration of omics data. Since the authors aim to be comprehensive, they needed to keep the description of the diverse modelling approaches superficial. But as such the review is a valuable primer for scientists that aim to enter the in silico world and to apply modelling to interpret large-scale omics data. The Metabolites journal will be a good platform to reach out to this reader.

The following are some criticisms the authors should address to improve the presented manuscript:

Major comments:

The description of examples from the literature often contains non-significant details. It would be more essential to emphasize the concepts used for instance to integrate omics data. The information in the text and in the table are also partially redundant. I recommend being more concise in the text and refer the reader to the original literature or Table 1 for details

Table 1 (I might have overlooked it: Do the authors refer to this table at all in the main text?): Instead of listing the publication titles, a concise summary of the major (computational) novelty used and the major findings made would be of more value for the reader.

Although very comprehensive, the manuscript is missing a few important developments in the field of metabolic modelling and omics data integration:

The very powerful ME models are mentioned in one sentence but not introduced to the reader. The same holds for the leaner variants of ME models, such as protein allocation models that have proven valuable for improving the predictions of stoichiometric models. Incorporation of kinetic parameters such as in the GECKO approach is another important development in stoichiometric modeling. Relevant literature includes: doi: 10.1038/nature24299, doi: 10.1038/srep36734, doi: 10.1101/2020.02.10.941294, doi:10.1371/journal.pcbi.1004913, doi:10.15252/msb.20167411, doi: 10.1016/j.cels.2017.11.013

Without mentioning machine learning techniques and the promising blending of machine learning approaches and mechanistic metabolic models this review will be incomplete and not representing the state-of-the-art. While a full overview of this field will be beyond the scope of the review, at least future directions including such (hybrid) modelling approaches should be discussed, especially with regard to omics data integration.

The importance of standardization in the formulation of metabolic models and in implementation of the computational approaches is mentioned in line 654. This important topic should be discussed in more detail either at the beginning of the manuscript are in the conclusion, as non-standardization is a big limitation in this field as recently discussed in doi: 10.1038/s41587-020-0446-y.

Minor comments:

l. 10/11: Biological systems ae controlled on the level of transcription, translation, posttranslational modification, allosteric effects. As such metabolic networks don’t operate autonomously from the other “layers” of control as the first sentence of the abstract suggests. Indeed, this statement is contrasted by the following sentence, in which the authors refer to the different mechanisms that impact the metabolism. This contradiction should be removed.

Line 16: It might not be clear to all readers what the authors mean with biochemical constituents at different layers. Please consider reformulating this sentence, e.g., by quantification of …constituents at systems level.

L. 20/21: constraint-based modelling and kinetic modelling (non-capitalized and modelling with double ‘L’ to be consistent)

L.29: “Cell metabolism gathers all the biochemical reactions occurring within the cell” ‘Gathers’ seems to be an inappropriate verb here. Consider replacing by ‘represents’.

L. 40: either remove “at different layers” are introduce the concept of dividing biological systems into different levels of organization.

L. 42: “development of the high-throughput technologies behind metabolomics” development of the high-throughput technologies for metabolomics

L. 52: by instead the so-called data-driven approaches

L. 60/61: This sentence needs improvement. Can functions be carried out by information? The second “that are” needs to be removed to link the following sub-clause to ‘cellular function’ instead of ‘levels of information’.

L. 70: reactions’ stoichiometry

L. 88: modelling modelling (or vice versa)

L. 109: the system’s stoichiometric matrix

L. 124/25: Reference missing.

L. 128: Corrupted formatting of equation

L. 135: ‘and’ after genome missing: methods to automatically annotate the genome and to derive….

L. 135: ‘It is achieved’: What does it refer to?

L. 145: FBA computes a specific flux distribution but it should be mentioned that this solution is not unique.

L. 160: add gene: gene knock-out

L. 160: TO make the sentence better comprehendible consider replacing with: ‘While FBA with the objective of maximizing growth results in reasonable solutions for wild-type cells, it does not so for (unevolved) gene knock-out mutants.’

L. 163: A reference for the MOMA approach is missing.

L. 170: present tense seems to be more appropriate: problems, which cannot be addressed…

L. 171: Not clear if the authors mean changing steady states, i.e., if the adjective should be linked to steady states as well.

L. 176 is instead or are (different way is..)

L. 181/82: replace ‘ in a constraint-based model’ with ‘under steady-state conditions’

L. 183: …that they (= the models) rely on’

L. 184: As the authors describe further below, single-point measurements of metabolite pool sizes can be integrated into CBMs as thermodynamic constraints that restrict the directionality of reactions. Consequently, this sentence needs to be removed are changed.

L. 186: not clear what the authors are meaning with “applicability of this approach”. Please improve sentence.

L. 188: allosteric, regulation, which depends…

L. 189: Appending CBMs with additional constraints is another approach to overcome limitations of FBA-based methods and might be included here (especially as the following sections are dealing with this approach).

L. 196: Could the authors be clearer what the difference between constraining extracellular reaction rates in FBA versus MFA is? For me this actually the same and I only see a difference in how the equation system is solved (linear programming vs least squares)

L. 197: constrains (s instead t)

L. 214: Why would genome-scale models need to be curated to be reduced to smalls-scale. Manual curation of genome-scale models should be done independent of reduction of the model for specific (e.g., MFA) applications

L. 215: Reference is missing for dynamic FBA

L. 218: secretion and uptake rates

L. 219: fluxes, which (comma missing)

L. 222: The number of metabolites, for which the labelling pattern can be determined is not more limited than in label-free metabolomics. The metabolites are not analyzed to the same extent as the information would be redundant, e.g., 13C-labeling patterns of metabolites of a linear pathway, in which the carbon backbone is not altered carry the same information, measuring allis hence redundant.

L. 224: The biochemistry of metabolic reactions is well understood and hence atom transitions are available for almost all reactions in a metabolic network, even at genome-scale. What is primarily limiting 13C-MFA is rather the resolvability of the fluxes with the information from 13C incorporation patterns.

L. 229: as proteinogenic amino acids are also bound entities, I suggest replacing “even” with “further”.

L. 230: the analysis (computation) of fluxes is increased to genome-scale, not the measurements.

L. 236: Kohlstedt and Wittmann used parallel ILE but didn’t employ 13C-labeled glutamine. A reference for the latter should be added.

L. 242: [42] seems to be the wrong reference as it doesn’t describe the integration of thermodynamic considerations in 13C-MFA.

L. 292: What is meant with ‘metabolism biology’?

L. 322: First sentence of figure legend not comprehendible (description of reviews omics approaches formalization techniques…)

L. 324: different layers of the hierarchical organization of biological systems

L. 331: Unclear what is meant with “amino acid rebalancing”. It is worthwhile explaining this ti the reader.

L. 407: They run (or performed metabolomics on different S. cerevisiae strains…

L. 425: A, B, C, D, are substrate (product) CONCNETRATIONS

L. 429: As the two sentence contradict each other, I recommend changing to: With the advent of the group contribution methods and the ability to compute Gibbs free energies for any metabolic reaction, this method can be applied on a genome-scale (or large-scale, system-wide, ..).

L. 446/ L. 448: don’t capitalize elementary flux modes

L. 480 specific antibody productivity

L. 481: label-free

L. 482: replace the indefinite ”it” with “this limitation”

L. 485: an alternative approach is to use

L. 486: “also” not necessary, consider deleting

L. 485-499: The information content of this paragraph is low. I recommend to condense the description of the presented study to the most important information, which is the use of an objective function that minimizes the number of exchange metabolites.

L. 504: Acetate secretion was predicted in the study in ref 94 because of an upper bound set on the oxygen uptake rate. This should either be mentioned or the emphasis rather be put on the simulation of the fermentation dynamics.

L. 508 I assume the authors mean that the approach smoothed the calculated fluxes not the calculation. Sentence needs to be changed accordingly.

L. 514: dynamic

L. 559: in a chemostat

L. 593: they integrated (past tense)

L. 594/5: add titles to improve readability

L. 595, “Besides, it …” what is “it” referring to? In general, I recommend avoiding the use of inexplicit ‘it’

L. 598: with different gene knockouts (notation of knockout differs and should be unified).

L. 615: constrain

L. 651: please explain the expression “forward problems” to the reader

Author Response

Response to comments made by Reviewer

Volkova et al provide a very comprehensive review on metabolic modelling and the integration of omics data.

We thank Reviewer 2 for the positive assessment of our work.

-The description of examples from the literature often contains non-significant details. It would be more essential to emphasize the concepts used for instance to integrate omics data. The information in the text and in the table are also partially redundant. I recommend being more concise in the text and refer the reader to the original literature or Table 1 for details. Table 1 (I might have overlooked it: Do the authors refer to this table at all in the main text?): Instead of listing the publication titles, a concise summary of the major (computational) novelty used and the major findings made would be of more value for the reader.

In general, we included details about the problem that researchers were solving, data that they used and what they have achieved in order to give perspectives to the readers of the possible research outcomes given the type of data and problem they have.

We corrected the Table 1 following reviewer’s suggestion. We would like not to disrupt the flow of the text and leave most information in the text. Table is included for quick read with links to original paper and main methods and data types used. Hence, text and table are complementary. We included more details in the table and removed the title column following your suggestion. Table 1 is referred to in the text in the Line 366-367: “. In this section we will discuss different data integration examples (summarized in Table 1).”

- The very powerful ME models are mentioned in one sentence but not introduced to the reader. The same holds for the leaner variants of ME models, such as protein allocation models that have proven valuable for improving the predictions of stoichiometric models. Incorporation of kinetic parameters such as in the GECKO approach is another important development in stoichiometric modeling.

As the reviewer correctly mentioned, ME-models and GECKO approach are very powerful, even though not straight-forwardly available for metabolomics data integration. Following the reviewer suggestion we have introduced reference to ME models and GECKO in the section dedicated to methods - Lines 211-215:”We also refer to other articles covering other powerful and widely used techniques such as GECKO method [7], a method that enhances constraint-based methods to account for enzymes maximum capacity by incorporating kinetic constants, metabolism and expression models(ME-models) [30]and protein allocation models [31–34].” In the conclusion and perspectives we added the information about GECKO approach and enhancement of constraint-based models with kinetic parameters Lines 883-892: “Accurate inference of large scale kinetic models from experimental data is the most promising technique, yet needs a significant amount of data points and efficient computational approaches as well as computational power. The combined application of both kinetic models and constraint-based models in the study of biological systems would allow to take advantage of their complementary features in a synergistic manner with the potential to overcome existing setbacks. The combination of the methods can be achieved in different ways, from modelling different parts of the metabolism with either kinetic modelling or constraint-based modelling [119,121](or other formalisms) to enhancement of constraint-based models with kinetic parameters information, such as in GECKO approach [7]. ”

-Without mentioning machine learning techniques and the promising blending of machine learning approaches and mechanistic metabolic models this review will be incomplete and not representing the state-of-the-art.

We appreciate the reviewer’s comment, with which we fully agree. The hybrid methods between machine learning and mechanistic modelling is the approach that many researchers consider to be the most promising areas to develop in metabolic modelling and omics integration. To address this comment we added text, where we mention the paper White-box modelling paper doi.org/10.1016/j.cell.2019.04.016 as an example of such hybrid approach. Lines 893-904 “Another hybrid approach is to combine mechanistic and statistical or machine learning models. It is challenging to identify the causalities and mechanisms by using machine learning approaches. Recently, the article showing “white-box” approach [148], as opposed to “black-box” uninterpretable machine learning models, was shown, demonstrating the power of such method. Modern machine learning algorithms are capable of pinpointing associations from datasets’ inputs and outputs but are not able to give mechanistic explanations of the phenomena. To address this challenge, researchers used an integrated screening-modeling-learning approach in which they linked input and output data with mechanistic models. White-box machine learning is potentially capable to identify biologically interpretable biomarkers, personalized treatment strategies and mechanisms of disease pathogenesis from big datasets.”

-The importance of standardization in the formulation of metabolic models and in implementation of the computational approaches is mentioned in line 654.

We appreciate this comment by the reviewer and definitely agree that this issue is one of the most important to overcome in order to progress faster as a scientific community. We added a paragraph with references about standardization and good practices in models and data sharing.

Lines 940-948. “Standardization of metabolic model formulation and simulation is an important task for researchers and the scientific community in order to advance the field of metabolic modelling and data integration in particular. Several initiatives and tools were proposed in order to harmonise the modelling studies, including MEMOTE (for metabolic model tests) [157]and The Systems Biology Markup Language (SBML) [158]. Also, good practices of models and data distribution should be more widely adopted in order to allow different scientific groups to integrate different datasets and use already published models. We refer the readers to the FAIR (Findability, Accessibility, Interoperability, and Reusability) guidelines [159].”

We appreciate thorough review and the numerous minor comments. We addressed all the comments following suggestions of the review.

Reviewer 3 Report

The manuscript provides a comprehensive summary of the many possible approaches to metabolic modelling and thus provides a useful resource for other researchers in the field. I would have preferred a more critical approach, with more comments on the usefulness and applicability of the methods. It would also have been useful if the authors could have given more indication of how the methods they describe have improved our understanding of the integration of metabolism. All of the methods are computationally practicable, but the return on the effort required to implement them is very variable. As researchers who use many of the methods routinely it would have been good if you could have given clearer recommendations of which methods to use for which purposes. Despite these criticisms, the article is well written and it provides helpful guide to the literature of metabolic modelling.

A minor point - please rephrase the first sentence in the legend for Figure 3 (it makes no sense as it stands).

Author Response

Response to comments made by Reviewer

-The manuscript provides a comprehensive summary of the many possible approaches to metabolic modelling and thus provides a useful resource for other researchers in the field

We thank reviewer for the positive assessment of our work.

- I would have preferred a more critical approach, with more comments on the usefulness and applicability of the methods. All of the methods are computationally practicable, but the return on the effort required to implement them is very variable. As researchers who use many of the methods routinely it would have been good if you could have given clearer recommendations of which methods to use for which purposes

We appreciate the reviewer's comment. The submitted article aims to gather the most recent and widely used computational methods in the field of computational metabolic modeling used for metabolomics integration. Despite sharing recommendations on what method must be used in a given situation is important and vital for education, we believe that this is out of the scope of this manuscript and more appropriate for an opinion article, detailed protocol or workshop. On the other hand each developed method aims to overcome different challenges associated with different cases of study and data types. Thus, we believe that encompassing all the possible scenarios should be addressed in a review manuscript or book chapter focused on this topic. However we agree that some general guidelines would improve the manuscript:

In conclusion and perspectives section we added the following text in order to sum up general recommendations about the integration approached:

Lines 860-874: “In this review we have discussed several techniques and examples to integrate metabolomics data in order to study cell metabolisms. In general kinetic models are more suitable for well known metabolic systems, which on the other hand reduce their scope to model of small and medium size. Kinetic models are also suitable to predict the state of the systems and they can deal with allostery and regulation. On the other hand, constaint-based models have the advantage of scaling well for large-scale problems, and they require little prior information. We refer to the specialized article for the details of usage of mechanistic models [147]. The type of data and the system under study also plays an important role when choosing integration techniques. For example, dynamic techniques (i.e. D-FBA) should be used when the overall metabolic behavior of a system is researched, that is not constant along the experiment. Another example is the use of labelling data when there is need of a more detailed description of intracellular fluxes. We have also seen that gathering time-series data can dramatically improve the possibilities in modelling and interpreting the data.”

We also cover the case when researchers should consider using dynamic methods (such as dFBA, dMFA) instead of FBA or stMFA. See lines 570-574: “When there are distinct metabolic states throughout the time-series experiment (e.g. exponential and steady-state phases during cultivation, behavior of the system before and after stress introduction, the adaptation to the stress), it is necessary to split the data into distinct phases and construct models separately to study the different metabolic behaviors in each phase ”

And also highlighted what researchers should consider when choosing between non-labelling and isotopic labelling experiments - lines 264-272: “This more detailed metabolic information allows 13C-MFA to infer metabolic reactions fluxes, which otherwise couldn’t be resolved, such as cycles, bifurcations or the source of NADPH production, among others. Thus, when such part of the metabolism is of interest, researchers should consider adding isotope labelling to their experiments. However it implies more complexity in terms of experimental design and data analysis [40,49].”

- It would also have been useful if the authors could have given more indication of how the methods they describe have improved our understanding of the integration of metabolism.

We appreciate this comment by the reviewer. In this review we have tried to describe the main scientific result from the article in order to give readers the examples of possible research outcomes from metabolomics integration into mechanistic models. However, we had to balance it out with not getting very specific into the research field and to keep it comprehensible for a wider audience. We added some references of the work from the field, highlighting the breakthroughs done by mechanistic models. Lines 853-859 :” For instance, mechanistic models are a promising tool for drug design and personalized medicine [91,143], are capable of predicting response to drug combination based on single drug data [146]. There are examples of how metabolic engineering tasks are guided by metabolic modeling for increased production of industrially relevant compounds [144,145]. Broadly, application of mechanistic models was very fruitful in metabolism research with high potential in the area of data integration.”

We also have such highlights in the text:

For example, lines 628-631: “Using time series metabolomics and exometabolomics, Bordbard et al [110]were able to predict the fate of citrate in red blood cells, which had previously been mostly neglected. This finding was also validated by isotopic labelling.” Here we refer that by integration of metabolomics data, researchers were able to discover new reactions taking places in the red blood cells. For the details of the research work, we refer to the original article. Or in this example, lines 649-659: “As an example, Christodoulou et al performed a time-series intra-cellular metabolomics experiment to study the metabolic response of E. coli to a peroxide pulse [112]. To further characterize the system they built multiple kinetic models to test for important allosteric regulators that could lead to such response. The authors tested thousands of combinations of allosteric regulators, and selected those that best described the metabolite concentration changes after the peroxide challenge. They found a number of relevant interactions that were further validated. Based on these findings the authors proposed that E. coli reserves the pentose phosphate pathway flux capacity during normal growth in order to perform a rapid response for oxidative stress.” The pinpoint of importance of specific allosteric regulation led to systemic understanding of regulation of central carbon metabolism in oxidative stress. This study further led to next articles that generalised this theory on different organisms.

-Minor comment

We changed the title of Figure 3.

Reviewer 4 Report

Comments to the Authors

Volkova S. et al. reported on the “Metabolic Modelling as a Framework for Metabolomics Data Integration and Analysis” (manuscript number: metabolites-836459). This author's review describes the current state-of-the-art in metabolic modeling and discusses future perspectives and current challenges in the field in detail.

Please correct only one minor comment.

Minor comment

In order to improve the accuracy and comprehensiveness of metabolic modeling, the development of metabolome measurement technology is also an important issue. For example, acquisition of quantitative metabolome data (accurate intracellular metabolite pool size) can contribute to improvement of model accuracy. Therefore, progress in a comprehensive metabolite measurement method using the stable isotope dilution method is desired. Identification of unknown metabolites is also an important issue in constructing an accurate metabolic model. Furthermore, the acquisition of quantitative and comprehensive information that takes into account the localization of metabolites in cells will be necessary for future studies of metabolic modeling (metabolite localization is discussed in the text).

Please briefly describe the above contents in the Conclusions and perspectives section.

Author Response

Response to comments made by Reviewer

-This author's review describes the current state-of-the-art in metabolic modeling and discusses future perspectives and current challenges in the field in detail.

We are thankful for the positive assessment by Reviewer 4.

-In order to improve the accuracy and comprehensiveness of metabolic modeling, the development of metabolome measurement technology is also an important issue.

We thank the reviewer for this suggestion. We added text in lines 904-924 to address this comment.

Lines 903-923 “The development of metabolome measurement technology also plays an important role in the accuracy and comprehensiveness of metabolic modeling. The most important directions are accurate quantitative acquisition, identification of unknown metabolites and subcellular metabolite localization. First, acquisition of quantitative metabolome data can contribute to improvement of model accuracy. Therefore, progress in a comprehensive metabolite measurement method using the stable isotope dilution method is desired [149]. Furthermore, identification of unknown metabolites is also an important issue in constructing an accurate metabolic model [150]. It can uncover shed lights on new reactions and better determine scope of the model. Metabolomics data integration dealing with different cell compartments remains a challenge. Existing experimental methods usually capture pooled metabolite concentrations. For example, for eukaryotic cells it is not possible to obtain the real concentration and/or isotopic labelling in different compartments from the pooled data. Both experimental and computational improvements are needed to solve this issue. So far, different experimental approaches have been proposed for sample preparation to extract mitochondria and leave the internal metabolites intact [151–154]. However, only mitochondria has been isolated for a labeled and label-free analysis of the mitochondrial metabolites. Therefore, studies are limited where researchers are focusing on metabolites that only exist in cytosol and/or mitochondria, while the metabolism in other cell compartments is neglected”

Round 2

Reviewer 1 Report

This can now be accepted. I found references to applications in drug discovery particularly important as this is where the applicability of such methods needs to be demonstrated.

Reviewer 2 Report

My comments have been sufficiently addressed.

Reviewer 3 Report

I am content for there to be a difference of opinion on the emphasis of the review, and the revised review provides a useful resource for other researchers in the field.

Please note that the manuscript has two different reference lists, one with 153 references and one with 159.

Reviewer 4 Report

My indicated point is corrected.

I think this version of the manuscript provides a wealth of useful information for Metabolites readers.

Therefore, your manuscript should be accepted in Metabolites.